# Hospitalisation at Home of Patients with COVID-19: A Qualitative Study of User Experiences

**DOI:** 10.3390/ijerph20021287

**Published:** 2023-01-10

**Authors:** Jose Cerdan de las Heras, Signe Lindgård Andersen, Sophie Matthies, Tatjana Vektorvna Sandreva, Caroline Klint Johannesen, Thyge Lynghøj Nielsen, Natascha Fuglebjerg, Daniel Catalan-Matamoros, Dorte Gilså Hansen, Thea K. Fischer

**Affiliations:** 1Department of Clinical Research, Copenhagen University Hospital—North Zealand, 3400 Hillerød, Denmark; 2Department of Respiratory Medicine and Infectious Diseases, Copenhagen University Hospital—North Zealand, 3400 Hillerød, Denmark; 3Department of Virology and Microbiological Special Diagnostics, Statens Serum Institut, 2300 Copenhagen, Denmark; 4Department of Communication, Madrid University Carlos III, 28903 Madrid, Spain; 5Institute of Public Health, Research Unit of General Practice, University of Southern Denmark, 5230 Odense, Denmark; 6Department of Public Health, University of Copenhagen, 1353 Copenhagen, Denmark

**Keywords:** COVID-19, home-based care, hospital at home, patient perspectives, telemedicine, user perspectives

## Abstract

Hospitalisation at Home (HaH) is a new model providing hospital-level care at home as a substitute for traditional care. Biometric monitoring and digital communication are crucial, but little is known about user perspectives. We aim to explore how in-patients with severe COVID-19 infection and clinicians engage with and experience communication and self-monitoring activities following the HaH model. A qualitative study based on semi-structured interviews of patients and clinicians participating in the early development phase of HaH were conducted. We interviewed eight clinicians and six patients. Five themes emerged from clinicians: (1) staff fear and concerns, (2) workflow, (3) virtual closeness, (4) patient relatives, and (5) future HaH models; four themes emerged from patients: (1) transition to home, (2) joint responsibility, (3) acceptability of technologies, and (4) relatives. Despite technical problems, both patients and clinicians were enthusiastic about the conceptual HaH idea. If appropriately introduced, treatment based on self-monitoring and remote communication was perceived acceptable for the patients; however, obtaining vitals at night was an overwhelming challenge. HaH is generally acceptable, perceived patient-centred, influencing routine clinical workflow, role and job satisfaction. Therefore, it calls for educational programs including more perspective than issues related to technical devices.

## 1. Introduction

A hospital-at-home (HaH) model provides hospital-level care at home as a substitute for traditional hospital care [1,2]. During the COVID-19 pandemic, hospitals all over the world have been challenged by an unprecedented number of acutely ill patients. Models to hospitalise patients with COVID-19 or future endemic disorders at home have been demanded [2]. A variety of models have been developed and tested [1]. However, our knowledge of the user perspectives of telemedical solutions for patient self-monitoring when hospitalised at home and patient–clinician communication at distance remain scarce.

The generally accepted definition of the HaH concept is the delivery of acute hospital-level care to patients at home [1]. It is an acute clinical service that takes all key services traditionally found in hospitals and delivers them to selected acutely ill people at home. It is episodic, comprehensively responsible for the episode, provides care 24/7, and includes medical, nursing, paramedical, therapy, laboratory, radiology, and pharmacy care [1,3]. Providing acute hospital-level care at home reduces costs, improves outcomes, and enhances the patient experience [1]. For patients acutely hospitalised due to COVID-19, improvements include reduced length of hospital stay and fewer readmission rates [4]. Following the first World HaH Congress in 2019, nine topics for future HaH research have been identified including patient and caregiver experience with HaH, technology, and telehealth for HaH, and education and training of HaH clinicians [1].

An HaH model is a complex intervention characterised by several interacting components that all need to be acceptable by users, and efficient and effective [5]. User perspectives are important to reach high quality healthcare; regarding HaH models, patients, relatives, and involved clinicians are together relevant.

Telemedical technologies have been presented as solutions to challenges of the provision of equitable, cost-effective, and efficient health services for more than two decades [6]. Much of the qualitative literature on telemedicine investigates chronically ill patients who are already familiar with their disease and used to cooperating with healthcare professionals and systems [7]. Prior to an acute situation, patients and relatives are already trained in and familiar with self-care at home and to take on increased responsibility for their own health. Many patients acutely ill with COVID-19 are significantly different from patients with chronic diseases [8]. Owing to the acuteness of their infection, symptoms, and the uncertainties regarding the development of the infection, their health literacy is generally lower, as is their familiarity with self-monitoring and the healthcare system per se [9].

We were engaged in the development and the early phase of evaluation of a telemedicine-supported HaH model. This study was performed with hospitalised patients who tested technology and workflows proposed for the HaH model without leaving the hospital. Based on a qualitative research design, this study aims to explore how patients hospitalised with COVID-19 and clinicians engage with and experience digital communication and self-monitoring activities as part of a proposed HaH model.

## 2. Materials and Methods

This study, conducted in Denmark, is part of a larger project named ’Hospitalisation of COVID-19 Patients at Home‘ between leading Nordic research groups in Denmark, Sweden, and Norway.

The empirical data were sampled by semi-structured interviews of participants in the early developmental phase of a virtual hospital-based HaH intervention, i.e., patients, clinicians who provided care for the patients, and clinicians of the investigator group responsible for the development and evaluation of the telemedicine technology and the clinical workflows utilised in HaH.

The reporting of the study follows the Standards for Reporting Qualitative Research (SRQR) [3] and includes the consolidated criteria for the Reporting Qualitative Research checklist, COREQ.

### 2.1. The Proposed Hospital at Home Model

The detailed description of the development and further evaluation of the proposed HaH model are described elsewhere. The overall aim was to shorten the in-hospital stay of adult patients admitted to in-hospital treatment due to an acute, contagious respiratory tract infection like COVID-19. The proposed HaH model includes telemedicine-supported self-monitoring and an out-going hospital-led healthcare team for domestic treatment and other interventions not applicable to telemedicine. No in-home biometric patient monitoring by third-party clinicians from municipality, family medicine etc., or relatives is planned.

Patients are expected to perform a range of self-measurements (blood pressure, temperature, pulse, etc.) using mobile medical equipment for domestic use. The data are manually sent to the hospital via an app. Additionally, patients regularly perform self-assessment using a short questionnaire. All data are assessed by clinicians at the hospital via a web-based telemedicine platform accessible at any computer at their regular workstations. In the case of signs of clinical deterioration or if data are not registered timely by the patient, an alarm warns the responsible clinician for further actions. Communication between patients and clinicians is supported by either video consultation or regular telephone call.

As an early step during the development process, the technology and workflows within the proposed HaH model were evaluated qualitatively (early phase of HaH model evaluation). Our focus was on the end-user perspectives of (1) patient self-monitoring and data reporting and (2) the digitalisation of the patient–clinician communication to the format of video-based consultations and phone calls. Later, a feasibility study is conducted to test the feasibility, patient safety, and patient and clinicians’ adherence to the final HaH model, defined and described as a complex intervention [10].

#### 2.1.1. The Hospital at Home Model Setup Used for the Present Study

This early phase of HaH model evaluation was completed as part of the in-hospital test, thus prior to the patients being sent home for an actual HaH experience. Thus, all testing took place in a controlled and safe hospital setting following the national COVID-19 isolation restrictions. Two rooms at the Department of Pulmonary and Infectious Diseases, Copenhagen University Hospital—North Zealand, were used. Patients stayed in a hospital ward with access to toilet and bathing facilities. The furnishing (hospital bed, bedside table, and armchair) and decor of the ward were kept as usual. In the case there were more patients occupying the same room, our HaH patient was shielded in the bed at the window. Thus, the patient was in a far from home-like, but sterile, hospital-like environment. Second, we had a small office, the Virtual Epidemic Centre, with a computer workstation for the HaH clinicians.

By protocol, it was intended that the patient–clinician interaction was virtual. However, in the case of IT failure or acute need of communication, clinicians could enter the patient’s room or the patient could leave their room to reach out for assistance if needed. The COVID-19 restrictions included social distancing and personal protective equipment, rigorous restrictions of patient mobility, and of relatives’ visiting. Furthermore, meals were served in the patient’s room.

Clinicians involved in this early model test were regular nurses and doctors working at the department. They were educated by the investigator group about the HaH setup including how to use technologies and inform patients. Thus, clinicians were at the same time responsible for providing both virtual and standard care for the patients at the department. Representatives from the investigator group (an engineer, two nurses, and two medical doctors) were present 24/7 at the department during the test phase to provide support for the clinicians.

Patient self-monitoring was scheduled by clinicians. During day and night, alarms supported the patient to remember measuring and reporting. Patients were asked to self-assess their well-being at the end of reporting. In the case of missing data, a nurse was warned by an alarm.

#### 2.1.2. Patients Participating in the Early Phase of HaH Model Evaluation

Adult patients admitted to the Department of Pulmonary and Infectious Diseases, Copenhagen University Hospital—North Zealand, Hillerød, were eligible for enrolment if they were Danish-speaking; had residency within the uptake area of the hospital; were self-reliant; in need for further hospital treatment of a confirmed COVID-19 infection; medically eligible to virtual HaH; and provided oral consent. Furthermore, the installation of the project app required NemID (a common log-in tool for Danish government websites, etc., provided to most of the Danish citizens). The test period was mid-November 2021 to the end of January 2022. The HaH capability was one patient at a time. A total of seven patients were recruited to the test, of which five completed participation until discharge to home (participation time: 21 to 72 h, mean 35 h). Two withdrew their consent and left the intervention for standard care.

#### 2.1.3. Use of Technology in the Early Phase of HaH Model Evaluation

The digital solution for data transmission (input by patients and retrieval for healthcare staff) had been developed at the hospital. It consisted of both a patient part: an app to be installed on the patient’s own device (mobile phone), and two parts for healthcare staffs’ receiving and acting upon patient data: a mobile phone for warning only and a web-based platform installed on the stationary computer placed in the Virtual Epidemic Centre for transmission and analysis of patient data. Video consultations with patients were provided by a national web-based system.

Patients were educated by a nurse member from the investigator group on how to self-monitor, report data, and communicate virtually. Both oral and written information about the early phase of HaH model evaluation was provided.

#### 2.1.4. Technology for Self-Monitoring and Data Entry by Patients

Three devices were delivered to patients for self-monitoring of five vital biometric signs: (1) the MightySat^®^Rx Finger Tip Pulse Oximeter from Masimo, Irvine, CA, USA (respiratory rate, oxygen saturation, and pulse rate); (2) the UA-651BLE Blood Pressure Monitor from A&D Medical, Ann Arbor, MI, USA (blood pressure); and (3) FastTemp^®^ digital thermometer from Universal Medical Supplies ApS, Farum, Denmark (rectal temperature).

The patients installed two healthcare apps on their smartphone, one for video consulting (a national platform for video consultations) and a study app. The latter had relatively few functions, including (1) an overview of planned self-monitoring; (2) an overview of planned video consultations with access links; (3) manual data entering of the biometric data and patient questionnaire; (4) automatic real-time transmission of data; and (5) an alarm to the nurse.

#### 2.1.5. Technology for Data Retrieval and Monitoring by Nurses and Clinicians’ Communication

For remote monitoring of patient data, a web-based platform was used. An inbuilt digital decision-support system indicated incoming data and prompted different levels of action (i.e., maximum time to action) or missing data entry on the mobile phone worn by the responsible nurse. She could interact with the patient by regular phone calling to further assess the situation. If relevant, a video consultation was performed for visual clinical assessment of the patient or, if severe clinical deterioration was suspected, medical doctor assistance could be launched. The mobile phone worn by the responsible nurse could not be used for individual text messages.

### 2.2. Empirical Data

#### 2.2.1. Interviews

To explore perceptions, expectations, and challenges as perceived by users, we used semi-structured interview techniques based on interview guides. A total of fourteen semistructured interviews were conducted with six patients (three male and three female, aged 36–61 years (mean 46), five treated for COVID-19 infection, and one for acute asthma), a 25-year-old daughter accompanying her father, two female nurses being the main responsible for the clinical treatment (34/65 years), and six clinicians from the investigator group (one male, five females, aged 30–61 years (mean 41), four doctors and two nurses, three with a Ph.D.), all deeply involved in the development of the HaH model and the digital devices. From November 2021 to February 2022, all interviews were conducted by an anthropologist experienced in qualitative interviewing (second author, S.L.A.). Interviews of clinicians were conducted in a quiet office at the department, while all patient interviews were conducted by phone (audio only) with the patient at home. Owing to COVID-19 restrictions, no group interviews were conducted, and face-to-face meetings with patients were changed to virtual meetings. Except for one patient interview, none besides the interviewer and informant were present. Predefined interview guides (for patients, HaH nurses, and project investigators, respectively) were developed primarily by S.M., S.L.A. (authors), and M.M.L. (nurse, member of the project group) based on existing literature, dialogue within the investigator group, and previous experiences. Key topics included the HaH concept, responsibility, own experiences, and thoughts about the future transfer of the model to the actual home, and perceived challenges, fear, and acceptance. Additional key topics to patients included instruction, managing tasks, and communication; and to clinicians: qualifications and workflow. Launched by brief small talk to establish a comfortable environment, the informants were encouraged to speak freely and honestly. S.L.A. introduced herself as an anthropologist not otherwise involved in the HaH model under study. She based the conversation on the predetermined open-ended questions and other issues emerging from the dialogue. The planned length of the interviews was up to 60 min. Patient interviews varied between 20 and 60 min (mean 40 min), interviews of clinicians between 50 and 90 min (mean 66 min). The interviews were audiotaped and transcribed verbatim by a trained research assistant. No repeat interviews were carried out, and transcripts were not returned to informants for comments. The interviewer’s notes from the interviews were not transcribed.

#### 2.2.2. Participant Recruitment Procedure and Consent

Because of the small number of patients with HaH experience, we did not set up any inclusion criteria but invited all HaH patients for an interview. Using a convenience sample technique, invitation to interview included all seven patients admitted to the HaH setup, the two HaH nurses, and the most influential of the investigators. Shortly before discharge from HaH, the patients were consecutively invited for an interview by one of the investigators. In the case of consent, which was given by everyone, S.L.A. was handed their name and phone number and contacted the patients by phone for further information and arrangement. Shortly before the completion of the test period, the clinicians were invited one by one for an interview by S.L.A. All eight persons invited agreed to participate.

All study participants had oral and written information about the study and gave written informed consent. Everyone was informed that participation was voluntary and that they could withdraw their consent at any time with no further consequences.

#### 2.2.3. Qualitative Approach and Research Paradigm

Our analytical approach was positioned within a subjective epistemiology. We used an interpretative methodology sensitive to individual meanings. An inductive approach was undertaken to explore user perspectives of primarily biometric self-monitoring and virtual communication as defined by the overall research question.

#### 2.2.4. The Process of Analysing and Reporting

After finishing the last interview, the empirical data were transcribed, coded, and structured manually. The interpretive analytical work was based on the six-step method for thematic analysis described by Braun and Clarke [11,12]. These steps included (1) familiarisation with the data by reading and rereading, (2) generation of initial codes, (3) searching for themes, (4) reviewing themes, (5) defining and naming the themes, and (6) producing the manuscript. The analytical steps were carried out as iterative processes. Initially, S.L.A., J.C.H., and D.G.H. read and reread the transcripts and started initial coding following an initial open inductive strategy. J.C.H. and D.G.H. continued coding and searching for themes. Reviewing and defining of overarching themes and subthemes were headed by D.G.H. and discussed with J.C.H. throughout the review phase. All authors gave substantial input during the process of producing the manuscript.

#### 2.2.5. Researcher Characteristics and Reflexivity

The interdisciplinary research team involved in the designing, conducting, and reporting of this study included clinicians and researchers at different career levels, all with an interest in the development and use of medical technologies. The team represented different educational backgrounds (medical science, public health, anthropology, physiotherapy, nursing, communication, and IT), expertise in relevant research methods (clinical trials, qualitative studies, media studies, and public health), and research topics (hospital at home, virology, clinical communication, digital communication, patient-centeredness, organisational models of care, and pulmonary nursing). During the analytical process, we repeatedly discussed how our professional positioning and theoretical standpoints might have influenced data generation and analysis. S.L.A., J.C.H., and D.G.H. all are experienced in qualitative research: S.L.A. as anthropologist, J.C.H. a physiotherapist, and D.G.H. a doctor and full-time researcher.

#### 2.2.6. Patient and Public Involvement

Neither patients nor the public were involved in or otherwise took part in the research design, analysis, or reporting of empirical data.

### 2.3. Ethical Considerations

All participants gave informed consent. All data were anonymised and stored in a secure place approved by the Danish Data Protection Agency (P-2021-345) and following the General Data Protection Regulation (GDPR). We followed the Declaration of Helsinki for research studies. According to the National Ethics Committee in Denmark, the Biomedical Research Ethics Committee System Act does not apply to qualitative studies (j.nr 21068140).

## 3. Results

The empirical data illustrated the pandemic context. Patients were at the hospital and pretending patients being at home confused participants from time to time.

Data from patients and clinicians were analysed in an iterative process. Codes and themes were constructed to frame and interpret the data, totally five themes from interviews of clinicians, four from patient interviews (Table 1 and Table 2). The theme ‘patient relatives’ emerged in both data sets. Depicted quotes to support the emergence of the five themes are extracted from an unpublished master file containing interview data. Each quote is referred to by a fraction, e.g., ‘2/39‘, representing the ID of the informant (numerator—2) and the position of the quote in the master file (denominator—39).

### 3.1. Interview of Clinicians

The clinicians all began with an overall goal of HaH for saving healthcare capacities to those in need, which appealed to them. However, as explored during the interview, uncertainties at several levels were substantial. Although many challenges were foreseen, others were surprising; HaH was generally referred to with great enthusiasm and engagement. Five themes emerged: (1) clinicians’ fear and concerns, (2) clinical workflow, (3) virtual closeness, (4) patient relatives, and (5) future HaH models.

#### 3.1.1. Clinicians’ Fear and Concerns

Fear and concerns elucidated during the interview with clinicians mainly related to ‘organisational issues’, ‘patient tasks and responsibility’, and ‘technical problems’. Patients in the HaH setting were perceived as a new group of patients, by many categorised as ‘our virtual patients’. Both nurses and doctors worried about norm agreements and were afraid of hidden overcrowding. So far, care of virtual patients was perceived as an add-on task (4/438) (Table 1). Some informants felt ready to care for more virtual than usual patients in the future. However, others underlined the unknown need for resources and anticipated higher complexity of problems among patients ‘left’ in the hospital (8/889; 13/2112).

Both nurses and doctors were concerned about their own and future colleagues’ skills for examination and treatment at distance (4/472,484, 2/277). The introduction and training program mainly covered technical issues. Some felt left in limbo—in doubt how to accommodate patients at distance (2/277). In line with usual practice, the doctors decide patient monitoring, while the nurse team is responsible for making it work. It bothered informants how to build a professional relationship with virtual patients. Worries mainly concerned patients never meeting face to face with health professionals, but also, as a doctor explained, how to support continuity of care across transmission to HaH.

The need for matching expectations regarding patient responsibility and tasks was highlighted (3/82). As an example, patients need to accept waking up at night to monitor, although it may seem uncomfortable. Predefined process diagrams were perceived as a kind of safety net for the patients in the case of missing data (2/39). However, patients took on the responsibility to various degrees and experiencing inadequate patient cases scared clinicians. By others, it was perceived as anxiety-provoking to rely on patients’ timely monitoring and responsibility of calling in case of exacerbation. Especially, the nurses felt on uncertain ground being responsible for new and unproven procedures out of their hands.

A high number of teething troubles with communication equipment bothered everyone (8/1095), and failures affected their professional self-respect negatively. The high number of alarms was perceived as very stressful by nurses, especially false alarms and those to be handled within five minutes. The poor integration of the project platform and the electronic health record was time-consuming. Several still did not consider the HaH concept and equipment ready or valuable (2/257). To diminish patient failures due to data feeding, one argued for Bluetooth systems.

#### 3.1.2. Clinical Workflow

The theme clinical workflow covered ‘task flexibility’ and ‘time and meaningfulness’. It had become clear how the HaH concept introduces a second group of patients with other needs of attention than usual in-house patients (8/1189). Simultaneously taking care of both the usual and virtual patients was perceived as confusing and perceived inappropriate for the future (5/1410). Running back and forth to the Virtual Epidemic Centre (office) when engaged with other patients stressed the nurses (13/1788, 1802). Several suggested a specialised nursing team dedicated to the virtual patients (5/1406). Prescheduling of ward-rounds intervened in doctors’ usual ‘freedom’. At the same time, fixed appointments were by nurses experienced as saving them small breathers. The importance of meaningfulness of new tasks was mentioned repeatedly. Especially, running for alarms, failures of technology, and systematic monitoring in line with algorithms but untailored the individual patient were referred to as meaningless and upsetting job satisfaction (8/1051, 1061).

#### 3.1.3. Virtual Closeness

Video-based communication was perceived both as a positive and negative alternative to the usual face-to-face communication in the hospital bedroom. First, clinicians were free from personal protective equipment, allowing patients to read body signals and to better hear (3/74, 12/2297). Especially for patients hospitalised for extraordinarily long periods, dialogue with a clinician not fully equipped was perceived crucial. Second, the dialogue felt more private, and clinicians imagined their attention being more undisturbed when the time was scheduled and the room was emptied for other clinicians and patients. Third, both nurses and doctors raised concerns about clinical examination strategies and health outcomes, especially the risk of missing out on patients’ symptoms and signs. Several reassured themselves that practising would show and also expose a likely need for focused training.

#### 3.1.4. Patient Relatives

The HaH model gives relatives the possibility of participating in ward rounds. Being at home will furthermore make it easier for relatives to be updated on the disease course and severity (5/1312). The clinicians did not predict a substantial role for the relatives. However, some raised awareness that unintendedly a role as private nurse or secretary could stress relatives to a degree making them sick (4/732). It was proposed to ask relatives to consent before patients transitioned to HaH. A discussion of how to prepare relatives as well as clinicians at hospitals and institutions for their new and still undefined roles was requested.

#### 3.1.5. Future HaH Models

Three themes emerged regarding future HaH models with patients being really at home: ‘expected goals’, ‘relevant patient groups’, and ‘paradigm shift’. Keeping hospital resources to those patients most in need was by all perceived the overarching goal of future HaH models. Secondary effects were improving patient safety, health, and well-being, along with responsibility for their own health and time for recovery when released from hospital norms and restrictions. Several patient groups were suggested for future models, such as chronic patients who could restabilise within a short time, dehydrated elderly, patients with COPD exacerbation, or patients in long-term intravenous antibiotic treatment. Future patients need to be stable with only low risk of acute treatment needs, but clearly in need of monitoring and professional contact 24/7, i.e., in need of hospitalisation if this model did not exist. Additionally, reverse models with special out-going facilities were considered. Elderly and others with low digital competence were to be excluded. One mentioned that, for safety reasons, exclusion criteria could be so restrictive that the patient volume becomes too low to reach break-even for costs and professional comfortability and competences.

Implementing HaH entails a fundamental paradigm shift regarding both patient and clinicians’ understanding of ‘being hospitalised’ (5/1568). Hospital treatment and care will be much more patient-centred than today (4/646), and the usual taking for granted of patient availability is challenged (10/2169). As concluded, a substantial rethinking of hospital routines and good clinical practice and communication is a premise of the future. It was recurrently mentioned how appropriateness of nurse and doctor professional skills cannot be taken for granted and include several competences beyond the digital (3/150, 5/1370). Specialisation for virtual nursing, continuous medical education, and revision of medical curricula were proposed.

### 3.2. Data from Patients

Experiences of previous hospitalisations were used by patients to frame their experiences and thoughts of the HaH model under study, as were their days at home following discharge. They described a strong wish to leave the hospital earlier if safely possible. The HaH model was thought of as innovative. It was generally found as an acceptable way to return home despite the persistence of symptoms that normally would have prolonged their hospital stay. Three patients explained how going home included an oxygen gas cylinder while still dependent on oxygen subsidy. No patients continued the HaH biometric measuring at home.

During the analytical process, four themes emerged: (1) transition to home, (2) joint responsibility, (3) acceptability of technologies, and (4) relatives.

#### 3.2.1. Transition to Home

Positive thoughts of being discharged home earlier than expected were motivated by a mixture of strong wishes for leaving the hospital setting and dreams of being at home in their well-known everyday surroundings. Understanding the theme ‘transition to home’ as ‘from hospitalised to socialised’ supported our understanding of patient thoughts and dreams.

Fatigued and socially isolated in the room and in bed fostered feelings of helplessness and dependence on busy clinicians (11/2554). Dressed in hospital clothes and surrounded by sick and complaining patients reinforced the feeling of being sick (14/2050). The unpredictable rhythm of days and nights made them feel in an awkwardly immobilised, waiting position, waiting for drink and food, ward round, clinical supervision, biometric measuring, etc. The processes of personal protective equipment taught them that, if not an emergency, asking for a clinical visit often included substantial waiting time due to dressing and busy schedules.

In contrast, being home made them feel free to move around inside, go to the toilet without disturbing, pour a glass of water, eat meals they liked when they liked, phone friends, switch the TV on and off as they pleased, and move out into the garden, etc. (6/602). It was explained how wearing daily clothes, sleeping in one’s own bed and being undisturbed by clinicians and roommates contrasted their time being hospitalised (11/2544). At home, it was even clearer how much they had missed it.

#### 3.2.2. Joint Responsibility

As compared to standard hospital care, the HaH model entailed the patients to take on the responsibility of biometric measuring, assessment and monitoring, tasks that were otherwise fully taken care of by clinicians (1/399). Beyond measuring, patients experienced how they kept an eye on time and schedules and prepared to be relaxed and thus ready for blood pressure measuring, etc. Some patients found ways to remember these duties, for example, by setting alarms. The HaH tasks made patients feel more in touch with the disease and better understand biometric figures. Thereby, they experienced a new appreciated feeling of control and responsibility for own health (9/1155, 1/431).

However, one explained, partly taking on the responsibility of biometric measuring potentially makes you feel guilty, e.g., in case a time slot passed, wrong figures were transferred, an exacerbation was overseen, etc. (1/441). In addition to failing with the technological elements, you would unnecessarily disturb a busy clinician. Owing to mental unawareness and physical challenges, not all patients may be ready for the concept, several patients explained. It was highly underlined that the model was ‘for the right patients—not all’. Patient consent and thus an opportunity to stay hospitalised as usual was by few perceived crucial for future HaH.

#### 3.2.3. Acceptability of Technology

Comments on the digital devices and processes to be tested primarily concerned the app, the alarms, and the need for individualised introduction and supervision. Generally, the patients concluded that the devices were easy to understand and use, ‘almost fault-tolerant’, although all referred to problems making it work and feeling unsure if they did it correctly (9/1137, 6/570, 14/1696). Regarding introduction, some appreciated time on their own to try out the measuring and reporting procedures, while others felt comfortable when reassured by the nurse, aside the first couple of times (14/1752).

Knowing that the app included an alarm for immediate contact with a clinician made patients feel comfortable, as did the automatic warning of the nurse in case data were skewed or missing (7/968, 11/2646). Being contacted in case of exacerbation gave patients the feeling of individual interest and empathy from the clinicians.

Being your own nurse during night-time was experienced as the most intrusive consequence of the HaH concept (11/2147, 14/1902). These scheduled duties were by some perceived as stupid and clashed with the daily practice of putting the phone on flight mode during night-time. Using the phone for work made it difficult and close to unacceptable for a few to break this practice to make the app work. However, when keeping in mind that the price could be staying hospitalised instead of going home to their own bedroom, these informants reassured that better matching of expectations during the introduction would make it acceptable for future patients. In contrast to usual hospitalisation, the scheduling of biometric measuring and ward rounds gave patients a much-appreciated feeling of freedom and ‘time on their own’ (11/2595).

Perceived technical problems with the digitalisation of communication included adaption on older mobiles, download of the secured app, missing internet connection, reverb on a noisy line, and an unaccustomed low resolution of the video.

One patient described herself as lazy by nature and therefore a little sceptical when introduced to the HaH concept (9/1105). However, summarising, the idea of switching to digital communication and taking on duties and responsibility to leave the hospital earlier than usual was accepted by all test patients as well as the daughter.

#### 3.2.4. Patient Relatives

To patients, it was perceived as less burdensome to their relatives being discharged earlier to home. Although the concomitant responsibilities and duties were 24/7, taking part in everyday activities without being dressed in hospital clothes made their health situation appear less dramatic (6/649). It was foreseen that older patients may need support for HaH activities at home, and as explained by one who had called her mother to stay, severely diseased people may require help to manage the oxygen supply, shopping, etc. (11/2493, 2612). No concerns for relatives or their competences were mentioned.

## 4. Discussion

This study, siloed in a secure in-hospital setting, contributes to the knowledge of user perspectives on the acceptability and the relational, organisational, and competence aspects of an HaH concept. Our findings illustrate that despite technical problems and errors, both patients and clinicians were enthusiastic about the conceptual HaH idea. Escaping from hospital routines to socialisation at home was very motivating for patients. They willingly engaged and learned to handle new technologies, thereby achieving self-control through self-care. If appropriately introduced, treatment based on self-monitoring and remote communication was perceived acceptable for the patients; however, obtaining vitals at night was an overwhelming challenge. The alarm system under test caused much stress, especially to nurses. Surprisingly, the video-based ward-rounds introduced flexibility, predictability, privacy, and virtual closeness to patients as well as clinicians. Concurrent care of the usual in-house and future remote patients, respectively, is a great challenge to workflow and professional identity and competences. The need for reshaping doctors’ and nurses’ work along with pre- and postgraduate training programs was underlined.

### 4.1. Study Strengths and Limitations

This study is based on a very early testing phase of user perspectives on digital technologies for patients discharged to continue their hospitalisation at home. A virtual monitoring ward was arranged for clinicians, but due to safety reasons, testing took place with patients still at the hospital. Owing to this design, several study limitations are to be considered. User experiences are not based on patient experiences of being at home in their everyday milieu with the activities, privacy, and concerns that distance to hospital may cause, neither were clinicians. It is therefore important to keep in mind that user experiences, to some degree, are based on envisages of future implementation.

We conducted semistructured interviews with both patients and clinicians to better understand the various perspectives on the HaH model. Social desirability is always a possible bias in interviews. Furthermore, some informants’ close involvement in development of the model may have biased the results towards an optimistic point of view. Built-in premises of the study design included that clinician perspectives were based on a small number of cases and strict inclusion criteria for the HaH setup left out weak and elderly patients. Mostly due to technical issues, only a small number of cases successfully experienced a video consultation. Challenges during transition to home and of being at home were scarcely elucidated. Consequently, this may hamper generalisation to wider patient populations, experienced clinicians, successfully transition to home, and fully matured digital technologies. The Medical Research Council Guidelines on complex interventions points out the importance of process evaluation to inform future development by understanding how interventions work [13]. Although with precautions to generalisation, our results focusing on how users engage and experience HaH are important for future modelling and implementation. Lastly, it may be underlined that evaluation of accuracy and error rates of devices, or patient care outcomes such as readmission rate, was out of scope.

### 4.2. Discussion of Results

A survey among attendees of the first World HaH Congress 2019 pointed to the growing variety of clinical delivery models that claim HaH status, and thereby the need for a consensus statement on the definition of HaH as a starting point for research [1]. Our informants exposed different interpretations of HaH by their various suggestions for relevant patient groups. Patients could be referred from in-house hospitalisation, nursing homes, or their own home, i.e., beyond shortening of hospital stays, the overall goals included protecting patients from being hospitalised at all by proactive home-based care. Our results thus underline that agreement of goals and what qualifies as HaH is important for research, but also highly important during implementation in clinical practice [1].

The COVID-19 pandemic forced healthcare systems worldwide to implement remote patient care [14]. A paradigmatic shift from in-hospital care to hospital-based care at remote clinics or at home was introduced rapidly. Our informants agreed on the need and own readiness for this new paradigm of healthcare. However, not all patients are suitable for HaH, and not all clinicians feel comfortable with these extensive changes. Several studies and review papers have pointed out the urgent need for adaptation of digital, clinical, and organisational competences. In line with a scoping review, our clinicians found an overweight of technical issues in the training offered, leaving out crucial issues like professional roles and competencies, relational and communicative challenges, shared responsibility of care, and other safety related issues [15]. Furthermore, focus on processes, communication, and organisational issues are important, not only to qualify knowledge, skills, and attitudes, but also to motivate, engage, and avoid burnout [16,17]. A multidimensional shift in training and clinical culture is crucial [16].

A narrative review from telepsychiatry found how clinicians have to adjust techniques to inquire, engage, communicate, and listen effectively [17]. Replacing usual ward-rounds with video-based consultations were by patients thought of as a positive consequence of HaH. This is in line with a Danish study from general practice showing how patients experienced being seen and heard and, furthermore, felt relieved of a pressure to present problems at once before rushing out quickly [18]. Privacy, interpersonal closeness, and the dedicated time-slots were mentioned by our informants. However, our empirical data were based on a small number of video consultations, and all patients were in the hospital setting. Previous studies have pointed out the importance of the contextual factors of place and surroundings [19]. The patient’s choice of place, lighting conditions, noise, and other persons in the room may have a direct impact on the doctor–patient interaction, the amount of information the patient wants to share, and how the doctor engages [20,21,22]. Future instructions to patients and clinicians may improve quality of care by guiding on such issues.

A survey of 257 oncology patients enrolled in a remote patient monitoring program (RPMP) due to COVID-19 showed that the three program pathways: onboarding, monitoring, and exiting, were all perceived crucial to patients to successfully engage them in the program [23]. Our informants only put words to onboarding and monitoring. For example, both patients and clinicians underlined the importance of carefully matching expectations during onboarding. By protocol, the HaH activities self-monitoring and daily communication between patient and clinicians ended automatically at discharge. We suggest that conceptualisation and focusing on all three steps including exiting are important for future model development and implementation.

Clinicians were much aware of new tasks and the influence of HaH on the workflow. Shifting from usual care of patients lying in the hospital beds to care of remote patients and vice versa were perceived as a great challenge to clinicians, especially the nurses. Caring of in-house patients was framed as ‘our real work’. Furthermore, the new tasks such as data retrieval and virtual communication were to a large degree seen as an add-on to their usual portfolio of clinical and administrative tasks. When HaH and remote patient monitoring programs become the new paradigm for delivery of care, patient volumes increase markedly [1]. Our study put focus on the need to timely address end users’ concerns about hidden overcrowding, workload, and clinical workflow. In addition to these perspectives on future organisation of HaH, some informants reflected on allocating specialised teams of professionals to this patient group and model of care. This is in line with the literature discussing the optimal work force to promote remote patient monitoring programs and take on the responsibility of HaH [16,23].

Caregivers have previously been reported to play an essential role when patients are discharged from hospital [1,24,25]. It was therefore surprising that both patients and professionals intended them only minor roles in future HaH. However, few of the professionals proposed discussions of future roles, caregiver readiness, and risk of harm if responsibilities become too stressful. Spouses to severely ill patients are balancing between managing their own needs and meeting the needs for support of the patient [26]. The extraordinary pandemic situation with drastic social restrictions during hospitalisation as well as in private homes may to some degree explain why discharge to home was perceived as a release to relatives and why they were only pretended a minor role in the future HaH.

### 4.3. Perspectives for Future Research and Practice

An HaH model is a complex intervention to test and implement. A series of studies will be required to progressively refine the design before embarking on a full-scale evaluation [27]. This study adds important knowledge to improve future modelling of crucial parts of the HaH. The general agreement that HaH is a paradigmatic shift in great favour of patients seems crucial for conceptual storytelling during implementation [28]. Furthermore, supporting clinicians in simultaneously taking care of remote HaH patients and usually hospitalised patients may positively influence implementation processes and professional role clarification and job satisfaction. Therefore, based on our results, we suggest future research to focus on the impact to clinical workflow of different organisational models, educational programs for clinicians, job satisfaction, and impact on relatives when transition takes place to private homes.

## 5. Conclusions

Our main learnings from interviews with patients and clinicians conducted during the early phase of testing for HaH, which was conducted in a secure hospital setting with remote patient monitoring and digital communication, included the following: HaH is generally accepted, is seen as patient-centred in contrast to usual care, affects clinical workflow and job satisfaction, and calls for educational programs that cover a wide range of topics beyond technical device-related issues. Last but not least, clinical activities and patient demands differ from standard hospital care.

## Figures and Tables

**Table 1 ijerph-20-01287-t001:** Quotes from the clinicians’ interviews.

Quote Number	Informant ID/Line	Statement
**Theme 1: Clinicians’ fear and concerns**		
1	4/438	Scared, what about my real job tasks, here, I do also have my usual tasks to do
2	8/889	I do see, it makes sense […] in the long run. My worries include, that the more patients we admit to HaH, the more complex are those left and the more care do they require
3	13/2112	The patients left will be more complex and require a lot of care
4	4/472,484	’See, touch, listen‘ is key of practice training of nurses and nurse assistants […] You need specific competence development focusing on how to be a remote nurse
5	2/277	I need to get used to that they (the patients) are not just on the other side of the door, that I cannot just… […] When washing a patient or mobilise one, or whatever you do, you notice […] there are so many things that you talk about and notice yourself, and. You will miss some things. […] You do not get information easily.
6	3/82	Matching of expectations with the patient […] consent to, that although it is inconvenient, they have to wake up at 4 o’clock in the morning to do the measuring, because we, as professionals, need the data
7	2/39	It is a considerable job. And there are many (COVID patients that may deteriorate fast) […] All these flow diagrams made in case we do not reach the patient, what then to do? […] I think a nice safety net has been established
8	8/1095	When transitioning a patient for HaH, you are dependent on—that technology works. What they (the nurses) have worried most about, I think, is if the patients are capable of monitoring properly.
9	2/257	When it works, and there are no more teething problems, I think, then they (the nurses) see it. Just now, I do not think they find it positive at all
10	8/857	The more experienced you become, the more relaxed you will be in your professional position. But I am sure, that when we start to include the first patients for HaH, then, staff will perceive the situation as very insecure
**Theme 2: Clinical workflow**		
11	8/1189	You need to respond to two different groups of patients, when attending shift
12	5/1410	With a patient in front of you, I think, it will be difficult to prioritise a remote patient, one that you can’t see, instead of the one in front of you who may be in pain or something else. There are some practical issues, I see Ole here, but I can’t see Maren who is alarming my phone
13	13/1788, 1802	It just means that a nurse taking care of patients, needs to drop everything to address an alarm within five minutes; a clinical situation that is not critical, professionally critical [...] I am concerned that we will then use resources for unnecessary tasks
14	5/1406	My hope is and has been throughout the project, that some clinicians will be dedicated to care for the virtual patients, and only take care of them
15	8/1051	The patient has to be tested after six hours, according to the algorithm. If you had been admitted in a normal way, they would have made an appointment with the patient about, yes but […] then ’I will look after you before you fall asleep, and then we do not need to follow up before tomorrow morning‘. So, it is all about, you have to dare to go beyond the actual wording in the manual.
**Theme 3: Virtual closeness**		
16	12/2297	Without being aware of it, we continuously use body language, when we are talking right now, when we are talking using FaceTime, we do see the other person’s face and decode how our words have an impact on the other person. That is not possible when you are all wrapped in the personal protective equipment. You cannot decode anything at all. You cannot use your normal decoding techniques to actually understand the other person and read how the other person receives your information. And we are facilitating that.
17	3/74	If it is a COVID-patient, you go there totally covered up in your personal protective equipment, the patient might not know if it is the doctor or the nurse. Who is visiting me? Who is asking me all these questions? Then I do believe, with respect to communication, that it can be an advantage, that you are not covered with a mask, glasses and other personal protective equipment, that you can see the person. There is also a concern on patients with hearing loss, who need to lip read, but that is not possible when we are wearing all the personal protective equipment.
**Theme 4: Patient relatives**		
18	5/1312	And then I do also believe […] maybe it is easier for the relatives because they can follow the disease trajectory and do not have to call the department all the time when they know we are busy.
19	4/732	Q: ‘What kind of position do you expect the relative to take in such a future?’ Hopefully not a too important position. Hopefully, they do not feel burdened by responsibility. In the beginning, this will happen, because they do not know the extent to which their resource can be used. Because it does not have to be like that […] the relative will get a breakdown, and suddenly have to visit the GP to get sleeping medication or be admitted to the hospital, because of stress symptoms.
**Theme 5: Future HaH models**		
20	5/1568	We have to change mentality among staff and patients, so it gets an integrated part of the definition of being admitted to hospital, that this actually also can happen at home.
21	4/646	HaH is more patient-centred. It is also a collaboration when patients are in the hospital conventionally, but in this case, it is on the hospital’s terms.
22	10/2169	Our access to the patient and the patient’s data it is great, and the patient’s access to us is also great […] It is a copy of the hospital function, which has moved to the patient’s home…There is professional accessibility, but there is no nursing accessibility.
23	3/150	If healthcare professionals, both doctors and nurses, are educated like 50 years ago, can it then be expected, that they can handle a digital patient? [...] We educate people to make patient evaluation with stethoscope and manual examinations, what do we then do, when the patients suddenly are presented on a screen instead of the physical meeting?
24	5/1370	Challenges with education of the staff—to make them feel safe in the process, and how are all the workflows, because it is an all-new way of thinking. To have a virtual work environment instead of a physical.

Quotes are sorted by themes identified through the analysis.

**Table 2 ijerph-20-01287-t002:** Quotes from the patients’ interviews.

Quote Number	Informant ID/Line	Statement
**Theme 1: Transition to home**		
1	11/2554	Then I need one to bring yogurt, and then I need one to do this and that, and, ah, you get so fed up with yourself […] that you are so helpless
2	14/2050	It is not the same walking 50 m down a corridor with many sick people as walking 50 m down my road where I can enjoy nature and so
3	11/2674	I don’t know what time the nurse comes and say ’we need to test now’
4	14/1962	I don’t feel well if I don’t get a good night’s sleep. I know that I couldn’t have slept all night if I needed to monitor and transfer the data, but, but when you lay in a hospital bedroom together with other that would like to see the TV, others that would go for a cigarette or drink coffee, and others having to pee and things like that, then, actually, there is never peace and calm
5	6/602	Nice not needing to be hospitalised, because I can walk around in my own home, […] go to the toilet, arh, and then lay down on the couch, look TV, and I can make me a cup of coffee in the kitchen, yeah, be together with my family. And then, generally, I think, it becomes most people better to be at home than laying in a hospital bed
6	11/2544	Here, at home, I am not disturbed by noises in the night, because someone is admitted or transferred to the intensive unit or… And more, there is no one in the corridor shouting ’Is there a nurse here, I am going to the toilet?’
**Theme 2: Joint responsibility**		
7	1/399	You have joint responsibility. You don’t just lay back and let things happen. You join from the beginning
8	9/1155	If you want to be healthy and go home again, then, then every positive indication that, no matter if it is the blood pressure or pulse or oxygen, that makes you happy, so… It may give you kind of a boost when you see values yourself instead of just having someone else coming by to tell you, yeah?
9	1/431	You are quicker up going […] because you return to home, you can walk more around and you are joint owner of your disease, if you can say so …
10	1/441	You need to take on responsibility. If you are not measuring, if you are not doing those tasks, then you must take on the responsibility yourself that you are not becoming better
**Theme 3: Acceptability of technologies**		
11	9/1137	It is a question about how far you are in your disease trajectory, and how cognitive well bright you feel and so. But, there are tasks that you need to remember. But, difficult, no it is not difficult, for sure!
12	6/570	They (the oximeters) are made for the nurses to read. To say, the number being 99 at my first monitoring looked like 66 to me (laughing). But, then I quickly saw, that, of course […] I just saw it upside down.
13	14/1696	I struggled a little with my blood pressure, because, I measured 3–4 times because showed different values, and therefore I doubted if I had been too unrestful, if I should sit or lay down […] So, I lay thinking whether the values I send gave the impression that I was more sick than I really were, and things like that
14	14/1752	I think, if I have had a training period long enough to convince me that from now, with no doubt, I can send the right values; then I would feel completely safe with it
15	7/968	I think the concept is really good. You can go home having ‘a digital on-call doctor right the corner’
16	11/2646	Well, I just need to be sure I’m having the oxygen I need. And in case I doubt, are getting nervous or feel that something is wrong, I can push a button knowing that I come through immediately.
17	11/2147	The only thing I was upset about was the night I should do the measuring; I hadn’t slept for two nights. […] and I have had a sleeping pill 1½-2 h before…
18	14/1902	If I think it is a problem, if I need to wake up at night to measure those values and I feel really exhausted and unconcentrated, then I prefer to be at the hospital
19	11/2596	*(talking about the pre-scheduled consultations).* Well, I think it was brilliant! Because, sometimes you have the idea to visit the toilet before talking with the doctor, but, when will it be, at 10:00 or 12:30 or? […]. When you have a time, you have to reach the digital waiting room and then wait for the doctor. You know then, you don’t need to sit waiting for hours
20	9/1105	Human being like us, we are lazy (!) so in the beginning I was just like, do I, oh, do I bother to participate? […] Since I decided to participate and learned what it implied, I was thrilled. Being at hospital when you start getting better means that you also begin to bore. [..] Keeping an eye on your own oxygen rate and all that, so it was damned funny. But, being scheduled every 4^th^ hour was somewhat, you know… (laugh)
**Theme 4: Relatives**		
21	6/644	I think it will mean a lot to the family […] Being at home they can see you all the time. You are not at a hospital, it seems less dramatic
22	11/2493	(talking about having the oxygen cylinder at home) I notice how I become, and then I ask my mother to go out and turn up, because I’m in trouble
23	11/2612	Yes, you do (become dependent on other people). But, fortunately, I have lots of nice neighbours who have all said to me ‘in case you need groceries or something else, just call’

Quotes are sorted by themes identified through the analysis.

## Data Availability

Data are available from the corresponding author after reasonable request.

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
