# Peer review of "Hospitalisation at Home of Patients with COVID-19: A Qualitative Study of User Experiences"

_ijerph, 2023, doi:10.3390/ijerph20021287_

Round 1
Reviewer 1 Report
My compliments to the Authors, who used a qualitative approach with a solid and integrated methodological basis. These results will form the basis of further studies (quali- and quantitative) on home hospitalization.
I suggest explaining the two tables in more detail (e.g the meaning of quote number and informant ID).
I think the small number of participants is a limitation of this study, but this has already been discussed by the authors themselves.
Author Response
Response to Reviewer 1 Comments
Point 1: My compliments to the Authors, who used a qualitative approach with a solid and integrated methodological basis. These results will form the basis of further studies (quali- and quantitative) on home hospitalization
Response 1: Dear Reviewer, thank you for your interest in our article and for the time you have used to review and give feedback and comments for improvement. It is great to know you see this article as the basis of future studies on home hospitalization as we do.
Point 2: I suggest explaining the two tables in more detail (e.g the meaning of quote number and informant ID).
Response 2: We agree, it is difficult for a reader to understand the column “Informant ID” for that we have added in “3. Results” the following text at the end of the first paragraph:
“Depicted quotes to support the emergence of the five themes are extracted from an unpublished masterfile containing interview data. Each quote is referred to by a fraction eg. [2/39] that is representing the ID of the informant (numerator - 2) and the position of the quote in the masterfile (denominator - 39).”.
We hope this explanation satisfies your requirements for improvement. Changes are in red in the manuscript.
Point 3: I think the small number of participants is a limitation of this study, but this has already been discussed by the authors themselves.
Response 3: , you are right, as you have read, this has already been discussed between us in the “4.1. Study strengths and limitations”. We understand you do not expect any changes here.
We hope now you can accept our answer and proposed changes in the document and that our manuscript can be published. Thank you in advance. The authors of the manuscript.
Reviewer 2 Report
The topics covered in the manuscript are very important nowadays. The authors indicate the superior role of the patient in the therapeutic process. They also point to problems related to patient care during COVID-19. The described model is the result of the work of people involved in patient care. They describe both the advantages and disadvantages of the model used. They are aware of the essence of its introduction and improvement. I am impressed by the accuracy of the description of individual aspects. However, I am concerned about all the conclusions drawn because the number of patients included in the study was very low. Therefore, the information can be presented as a case study rather than a meta-analysis.
Author Response
Response to Reviewer 2 Comments
Point 1: The topics covered in the manuscript are very important nowadays. The authors indicate the superior role of the patient in the therapeutic process. They also point to problems related to patient care during COVID-19. The described model is the result of the work of people involved in patient care. They describe both the advantages and disadvantages of the model used. They are aware of the essence of its introduction and improvement. I am impressed by the accuracy of the description of individual aspects.
Response 1: Dear Reviewer, thank you for your interest in our article and for the time you have used to review and give feedback and comments for improvement. It is great to know that you are impressed by the accuracy of the description of individual aspects as we do. We have a really good team of professional researchers who can extract valuable information from participants via interviews. Thank you.
Point 2: However, I am concerned about all the conclusions drawn because the number of patients included in the study was very low. Therefore, the information can be presented as a case study rather than a meta-analysis.
Response 2: We are a bit confused here as our study is not a meta-analysis. This is not expressed in any part of the manuscript. At the same time, we understand that such kinds of studies are quantitative, formal, epidemiological studies designed to systematically assess the results of previous research to derive conclusions about that body of research. Typically, but not necessarily, such kinds of studies are based on randomized, controlled clinical trials.
In our view, our study is not compatible with a “case study” either, as we understand such a study to be defined as an intensive study about a person, a group of people or a unit, which is aimed to generalize over several units’. A case study can also be described as an intensive, systematic investigation of a single individual, group, community or some other unit in which the researcher examines in-depth data relating to several variables.
Instead, we consider our study as a qualitative interview study that is not following any variables but that just extracts users' perceptions and beliefs on the Hospital at Home model. We think as well we explain the limitations of our study clearly, pointing to the fact that the number of study participants is limited which indeed may hamper generalisation. Hoping that you agree with these considerations, we have improved our manuscript in accordance with the specific suggestions provided by reviewer 1 and hope that our manuscript can be published. Thank you in advance. The authors of the manuscript.